# A versatile and customizable low-cost 3D-printed open standard for microscopic imaging

Benedict Diederich [1,2,5✉], René Lachmann [1,3,5], Swen Carlstedt[4], Barbora Marsikova [1,3], Haoran Wang[1], Xavier Uwurukundo[1], Alexander S. Mosig [4] & Rainer Heintzmann[1,2,3]

Modern microscopes used for biological imaging often present themselves as black boxes whose precise operating principle remains unknown, and whose optical resolution and price seem to be in inverse proportion to each other. With UC2 (You. See. Too.) we present a low-cost, 3D-printed, open-source, modular microscopy toolbox and demonstrate its versatility by realizing a complete microscope development cycle from concept to experimental phase. The self-contained incubator-enclosed brightfield microscope monitors monocyte to macrophage cell differentiation for seven days at cellular resolution level (e.g. $2\,\mu m$). Furthermore, by including very few additional components, the geometry is transferred into a 400 Euro light sheet fluorescence microscope for volumetric observations of a transgenic Zebrafish expressing green fluorescent protein (GFP). With this, we aim to establish an open standard in optics to facilitate interfacing with various complementary platforms. By making the content and comprehensive documentation publicly available, the systems presented here lend themselves to easy and straightforward replications, modifications, and extensions.

[1] Leibniz Institute of Photonic Technology, Albert-Einstein-Straβe 9, 07745 Jena, Germany. [2] Institute of Physical Chemistry and Abbe Center of Photonics, Helmholtzweg 4, Friedrich-Schiller-University, Jena, Germany. [3] Faculty of Physics and Astronomy, Friedrich-Schiller-University, Jena, Germany. [4] Jena University Hospital, Institute of Biochemistry II, Am Klinikum 1, Jena, Germany. [5] These authors contributed equally: Benedict Diederich, René Lachmann. ✉email: benedictdied@gmail.com

Growing demand in biological research for spatial and temporal resolution, imaging volume, molecular specificity, and high throughput leads to ever more complex and expensive microscopes[1,2]. Alongside numerous imaging modalities, long-term observations of living organisms, which have minimal impact on their natural behavior, became an important aspect in light microscopy. The need to keep the cells in a well-controlled environment poses additional constraints being addressed by imaging inside an incubator[3,4] or exploiting on-microscope incubator units[5–8]. Assembling, maintaining and improving microscopes, as well as analyzing and verifying the produced data very often requires a consulting specialists dedicated to the respective instrument, thus further separating microscope engineers from their users[9,10]. For a large variety of imaging tasks, such as those mentioned above, tailored solutions are indeed commercially available, yet they are often costly, hard to extend or modify and rarely documented sufficiently to enable users adapting them for "out-of-the-box tasks", outside the range of their primary purposes.

Separately, in light of the growing pressure to publish as soon as possible, science is approaching a reproducibility and quality crisis[11]. Open research, in which every step is recorded transparently and made fully accessible to the general public, can help to restore the confidence in scientific literature, which has been visibly compromised in recent years[12].

Modern optical setups are reaching immense complexity, combining a growing number of optical and photomechanical components. They typically originate from different manufacturers adhering to various industry standards such as the International Organization for Standardization (ISO) or Royal Microscopy Society (RMS), whose intra-compatibility is often not guaranteed. This makes it particularly hard to tailor or even reconfigure optical systems, requiring handcrafted adapters or unnecessarily long attachments compromising the systems' integrity and stability.

What we see as a substantial space for improvement, is an open standard[13] permitting straight-forward interfacing between constituents of modern microscopes including sources, optics, optomechanics, and detector components. Such platform would facilitate simple constructions of versatile imaging instruments, easy to adapt to almost any imaging task at hand. The change from one imaging system to another could thus be reduced to a mere reconfiguration rather than a new design. Such a tool would be useful not only for research, but also immensely helpful in optics education. It would substantially reduce the effort required to build a setup and allow students to actively perform system reconfigurations within minutes. Such hands-on experience would lead to understanding and enabling everyone to perceive optics as a playground, where many ideas can easily be explored. In order to realize such a system, an open standard is paramount, as only in this way an effortless reconfiguration can be permitted without being overly restrictive to the possibilities. Luckily many great steps in this direction have already been made.

Recent approaches like the Flamingo[9,10] set out to establish light sheet microscopy-as-a-service to everyone, thereby addressing the issue of accessibility. A number of very well documented open-source projects such as the lattice light sheet[14] or openSPIM[15] nowadays spawn educational workshops and thereby attract users to contribute to its development. In terms of hardware design, projects like the "open-flexure stage"[16], the "100 € lab"[17], the smartphone-based "Foldscope"[18] and open-source single-molecule localization microscopy (SMLM) systems[19,20] demonstrate flexible and low-cost microscopy solutions capable of great performance. More generic approaches have been realized in the form of an opto-mechanical toolbox[21] and in form of a functional unit box-like approach called μ Cube[22]. With the

widespread availability of and easy access to rapid prototyping tools such as 3D printing, programmable electronics (e.g. Arduino[23]), high-quality cameras in smartphones or mini-computers (Raspberry Pi[24]), it is now indeed possible to develop an open standard that is accessible to everyone, thus ensuring wide dissemination, adaptation and expansion. Impairments to image quality due to less corrected inexpensive optical components or less stable mechanical arrangements, can often be real-time compensated by smart electronics and software algorithms. Methods like autofocus-routines, deconvolution[25], or the recovery of hidden information like the quantitative phase using simple LED arrays[26] are recent examples of such possibilities.

With our UC2 (You. See. Too.) approach, we strive to create such open standards. Relying on the concepts of matching focal planes (Fig. 1a) makes UC2 particularly easy to use, flexible to reconfigure and versatile for a large range of applications. It is equipped with open-source software, open design-files, and blueprints for a large variety of setups and openly accessible documentation. UC2 facilitates a cost- and time-efficient opportunity for pupils and students at all levels to experience designing and applying a variety of complex optical setups. It further enables access to modern light microscopy for a wide-spread group of users and developers by exploiting purely off-the-shelf consumer-available components (Supplementary Notes 1 and 8 for the bill of material) and thereby creating inexpensive microscopic imaging devices for around 100–400 Euro.

The manuscript details the entire development cycle of an incubator-enclosed bright-field microscope from its assembly to the successful application, where four identical systems are exploited to a parallel 168 h long imaging session of monocyte to macrophages in-vitro. The device is further transformed into a light sheet microscope, which exploits the original bright-field microscope assembly and only a few additional components. In order to demonstrate UC2's applicability to biomedical research, we provide imaging results from a variety of biological samples including fluorescing transgenic human pulmonary microvascular endothelial cells, *Drosophila melanogaster*, zebrafish, *E. coli* bacteria, which have been obtained using a range of UC2 based microscope modes, particularly the bright-field, wide-field fluorescence, image scanning microscopy, intensity diffraction tomography, and structured illumination.

## Results

**Open-Standard: The Basic Cube**. Modern microscopes with infinity-corrected objective lenses often follow the so-called 4f-configuration (Fig. 1a), where lenses are aligned in a way that focal-planes (f) of adjacent elements coincide to limit the amount of optical aberrations, to realize tele-centricity, and to predict the system behavior using Fourier-optics[27]. The name 4f results from the sum of the focal-distances of a simple imaging system with two adjacent lenses stacked with coinciding focal planes, leading to 2f per lens, hence 4f in total. We adapt this inherently modular design with a generic 3D-printable framework, in which individual modules (i.e. optical building blocks) in the form of cubes Fig. 1b and Supplementary Notes 2 are arranged in such a way that the focal planes of optics in successive cubes often coincide.

By analyzing many available optical components, imaging systems, and frameworks, we found that a design pitch of $d_{block} = 50$ mm seems to optimally balance compatibility, handling, and flexibility for enabling Fourier-Optical (4f) setups. Separating the cube into a base and a lid simplifies printing using standard fused deposition modeling (FDM) 3D-printers and allows to easily insert components as plug-ins.

Having neodymium ball magnets ($\varnothing_{magnet} = 5$ mm) positioned in a grid pattern on an extendable baseplate and ferro-

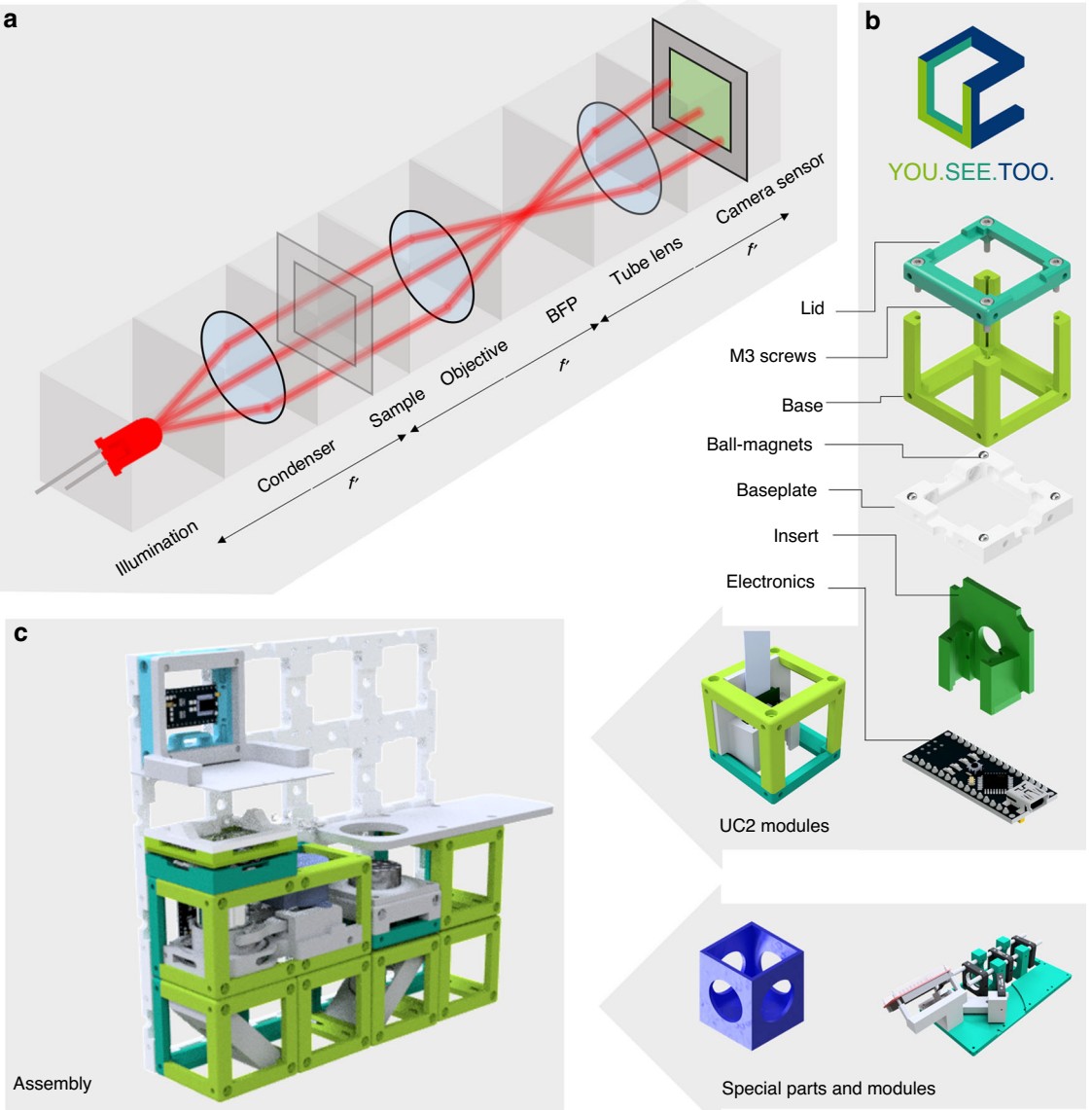

**Fig. 1 Optical Setup using UC2 Optical Building Blocks. a** The 4*f*-system divides Fourier-optical arrangements into functional units, where *f'* corresponds to the focal-lengths. BFP corresponds to the back focal plane (i.e. pupil plane). **b** The unit element (cube) acts as a base framework for any component which fits inside (lens, camera, Z-autofocusing mechanism, etc.). **b** A magnetic snap-fit mechanism connects the optical building blocks to a skeleton to realize mechanical stability and rapid-prototyping of a given optical setup. **c** An exemplary setup of a microscope for an ordinary smartphone (not shown) and an inexpensive objective as a combination of available modules. The cubes fit on the baseplate grid at 50 × 50 mm² design pitch (see Supplementary Notes 3).

magnetic cylindrical bolt screws (DIN 912) sitting in the cube's edges allows a stable and precise magnetic mount. Multiple orientations of baseplates allow to built in three dimensions. We found a four-point fixation as a good compromise between the common rectangular arrangement of optical setups and mechanical stability.

External electro and optical components (e.g. lenses, mirrors, LEDs; see Fig. 1b) and already existing equipment (e.g. rail-systems from Thorlabs, Quioptics, Edmund Optics) can be easily adapted by plug- and modifiable inserts (see Supplementary Notes 4). A module developer kit (MDK, Supplementary Notes 1) with a generic reference design for customized inserts provides a simple interface to work or add designs to the toolbox, even for users lacking technical training.

Scaling complexity of optical systems starting from a simple magnifying glass up to a fully working light sheet fluorescence

microscope (Fig. 2) is ensured by relying on the previously introduced library of modules that are combined and put in the appropriate order (Supplementary Notes 5). Adding more advanced consumer electronics (cameras, motors, video-projects, etc.) allows the use as smart microscopes and enables remote control. Micro-controllers ensure wired (i.e. $I^2C$[28]) or wireless (i.e. WiFi, IoT-based protocol using Message Queuing Telemetry Transport (MQTT)[29]) communication interface to trigger light-settings or focusing mechanisms (Supplementary Notes 6.1). Power is supplied through the conducting magnets or wires with rectifiers in the cubes.

**Versatile: a bright-field microscope for long-term incubator-enclosed in-vitro imaging.** The development cycle of creating a microscope visualized in Fig. 2 starts by identifying a problem which requires optical imaging. Here it is the minimization of

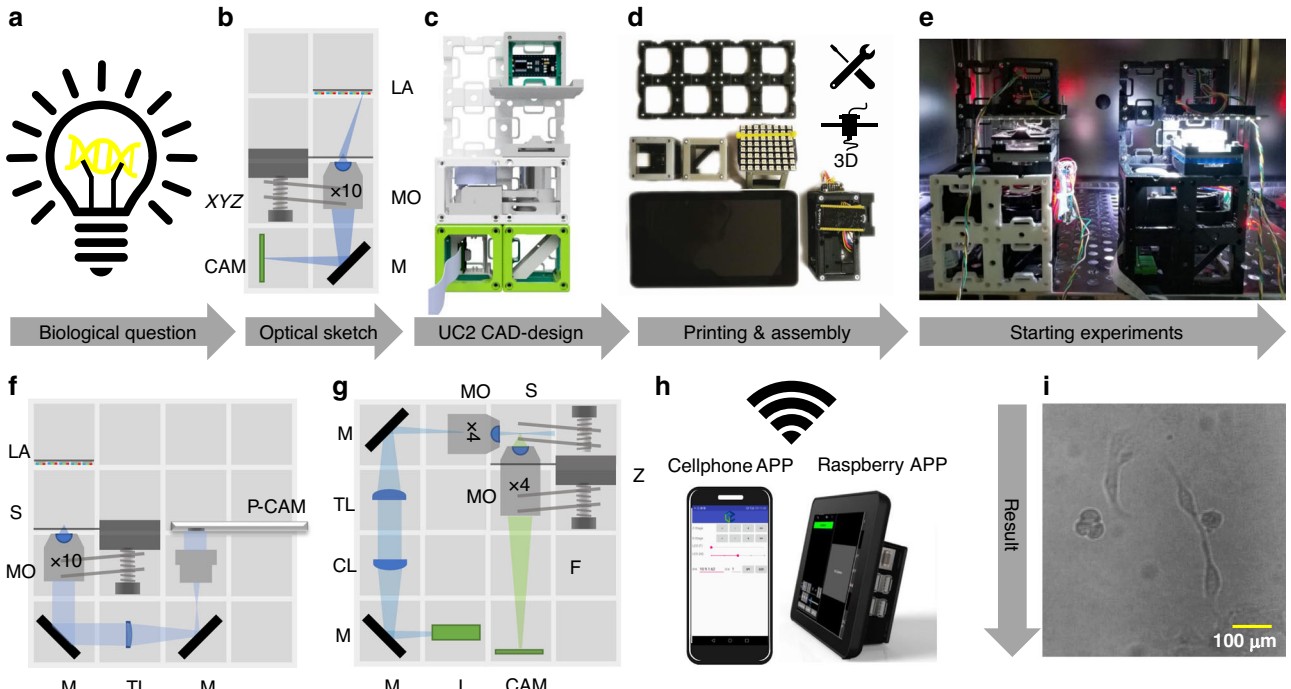

**Fig. 2 Rapid Prototyping using UC2.** Common workflow to build a UC2 application: (**a**) Starting with a biological question/idea in need of an imaging device drafted in (**b**) (inverted incubator microscope) and transferred using UC2 components from the CAD library in (**c**). After printing and assembling it (**d**) the device will be placed in its working environment (e.g. incubator) (**e**) ready to acquire long-term image series of e.g. MDCK cells visualized in (**i**) and Supplementary Video 6. Remote-control is granted using "smart components" (e.g. cellphone, Raspberry Pi) in (**h**). Reusing components allows the conversion into a cellphone-microscope (**f**) or light sheet microscope (**g**) within minutes (see Supplementary Video 5) and Supplementary Notes 7.8). CL: cylindrical lens, TL: tube lens, L: Laser, LA: LED-array, M: Mirror, MO: Microscope Objective, P-CAM: Detector (smartphone or Raspberry Pi), S: sample positioning stage, F: emission filter, Z: focusing stage.

external influences causing problems such as bacterial infections in in-vitro-experiments (Fig. 2a) of eukaryotic cells. We found that a small inverted microscope (Fig. 2b) in transmission bright-field-mode (BF) with an optical resolution on the subcellular level (i.e. <2.2 μm) for ≈300 Euro fulfills the quality requirements for long-term monitoring of human primary cell cultures. After combining the UC2-basic cubes, base-plates, inserts, and necessary components digitally to test for spatial limitations (Fig. 2c) we 3D-printed and assembled the system (Fig. 2d). For cross-verification, stability measurements and to increase the throughput, we placed four BF-setups (2× $I^2C$-, 2× MQTT-interface; two of them shown in Fig. 2e) into a single incubator. We designed a graphical user interface (GUI) on the Raspberry Pi, to preview the region-of-interest, set the imaging parameters (focus, illumination) and ensure autonomous image acquisition (Supplementary Notes 6.1).

We performed multiple long-term measurements under conditions of high humidity ( ≈100%) and at ≈37 °C, $CO_2 = 5\%$ over 7 days taking images at a rate of 1 frame per minute. This way we are continuously monitoring the morphological changes and plasticity during monocyte to macrophage differentiation (see Fig. 3a and b). As part of the innate immune system[30], macrophages notably reside within the tissue, where they act as phagocytosing cells involved in the clearance of pathogens and dead cells[31]. Monocytes can be isolated and differentiated in-vitro within 7 days in the presence of granulocyte-macrophage colony-stimulating factor (GM-CSF) and macrophage colony-stimulating factor (M-CSF) to macrophages. During the differentiation process, these cells increase in size[32] and are able to change their morphology depending on their polarization[33–35].

We cultured monocytes in 3 ml X-Vivo medium within 35 mm dishes. The shape of adhesive monocytes appears round and elongates during cell movement (not further quantified). The graph in Fig. 3d shows the increase in the area of individual macrophages over time. We observed a significant increase of size within 5000 min observation (Analysis Of Variance, NOVA with post-hoc Turkey's) in agreement with published reports[32]. Macrophage locomotion and phagocytosis is mediated by the concerted formation of pseudopodia. We were able to monitor pseudopodia formation[36] and to follow macrophage movement and associated morphological alterations in cell shape. This enables us to relate the elongated form of the macrophage to its movement (see Supplementary Fig. 1), which is increased upon detection of pathogens, damage associated molecular pattern, or cytokines[37]. We also observed phagocytosis (Supplementary Fig. 1) of dying and dead cells Fig. 2).

During imaging, the magnetically-mechanically fixed in-vitro sample (e.g. ∅ = 35 mm petri-dish, organ-on-a-chip, or standard microfluidic chips, e.g. Ibidi μ-chip) experienced a significant focus drift due to temperature-dependent deformation (see Supplementary Video 1), especially using Poly-Lactic Acid (PLA, Supplementary Notes 7.2). We found that Acrylnitril Butadien Styrol (ABS) outperformed PLA in terms of stability at higher temperatures. Even though a working autofocus routine (see Supplementary Notes 7.3) was developed, our ABS-printed stages proved sufficiently stable after a thermal equilibration period and so we decided to conduct our long-term incubator-enclosed experiments without the use of autofocus (see Support. Videos 1/2). A temporal drift analysis of the PLA printed stage is presented in Supplementary Notes in 7.2.

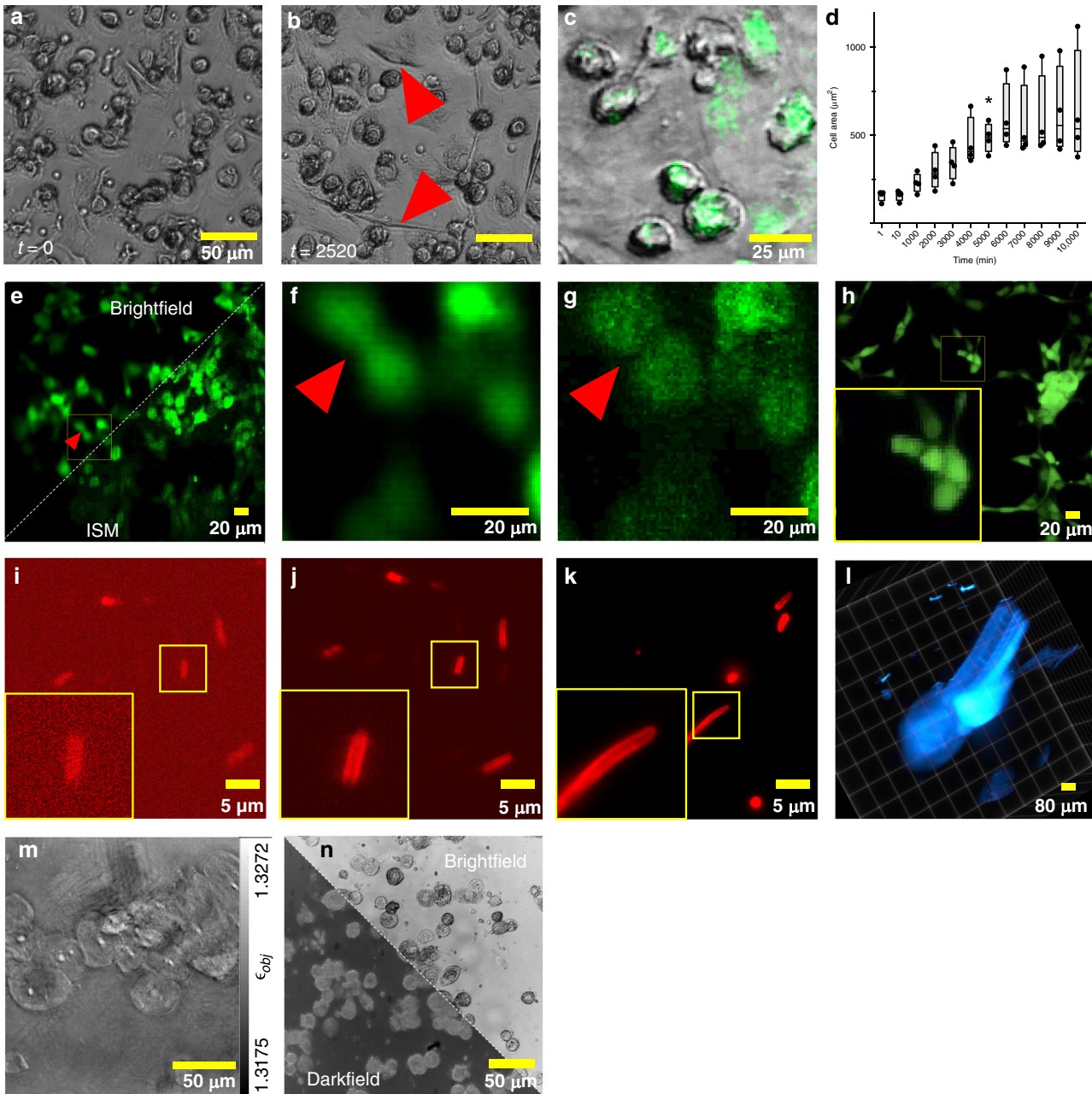

**Fig. 3 UC2 imaging modalities. a, b** Variation in macrophage's morphology, where elongated cells are clearly visible after 42 h (red arrow) imaged in transmission mode. **c** The bright-field channel superposed with a fluorescent signal of fixed macrophages labeled with CellTracker green captured with the incubator-enclosed microscope. **d** the growth of a differentiating cell is plotted as the average area of cells across multiple time-steps and different experiments ($n = 4$). Whisker plots: $10^{th} - 90^{th}$ percentile, the box represents the $25^{th}$ and $75^{th}$ percentile with the line in the box marking the median. Statistical testing with one-way ANOVA and Tukey's correction with GraphPad Prism (GraphPad, CA, USA), $p = 0.034$ ($F = 10.76$, $DF = 11$). Data of four independent experiments is shown. **e** Wide-field fluorescence (top-left) and the computed "superconfocal" result (bottom-right) of GFP-labeled Human Pulmonary Microvascular Endothelial Cells (HPMCs) illuminated with a laser-scanning projector, recorded with a cellphone camera. The zoomed-in images show the improvement of the optical sectioning in the case of structured illumination (**g**) compared to wide-field (**f**), where smaller cell-structures are lost. **h** A comparison of the same sample acquired with a commercial laser-scanning confocal microscope. A benchmark from the infinity-corrected fluorescence microscope using the Raspberry Pi (**i**) and cellphone camera (**j**) and a research-grade microscope (**k**) of mCLING-ATTO 647N labeled fixed *E. coli* bacteria, where the cellphone clearly resolved the bacterial membrane. **l** A Z-stack of a GFP-expressing zebrafish acquired with the UC2-light sheet. **m** Using an LED-ring as the illumination enables quantitative phase imaging of cheek cells using annular Intensity Diffraction Tomography (aIDT). **n** LED matrices can rapidly switch between bright- and dark-field imaging as shown in (**n**).

**Enabling: light sheet microscope for educational areas**. In this section, we demonstrate versatility by transforming the BF-system of the previous chapter into a light sheet microscope (Fig. 2g) by exchanging the LED array in favor of a laser-pointer, adding a second microscope-objective, beam-expander,

cylindrical lens, and the sample stage using a larger base plate. A video explaining the conversion together with a detailed conversion recipe and a detailed scheme of the open SPIM-inspired[15] setup is given in Supplementary Video 4) and Supplementary Notes 7.7 respectively. We acquired a 3D data-stack of zebrafish

larva expressing GFP in the blood vessels which was further drift-corrected and deconvolved using the "GenericDeconvolution" program by Heintzmann et al. (available upon request) (Fig. 3l, Supplementary Fig. 3). At the moment the results show merely a proof of concept that it is possible to build a light sheet system at such a low price (400 Euro). Better optical components and better adaptation to applications would be necessary for better performance. However, our light sheet microscope proved its usefulness in the educational area, giving the users a valuable insight into a method they frequently work with but know it only as a black box. We analyzed the minimum required number of printed and off-the-shelf components to build the formerly mentioned setups as well as telescopes, projectors, Abbe diffraction experiments, or holographic (e.g. lens-less) imaging devices in a cost- and resource-effective way to compile a ready-to-print collection of open-sourced parts and documentation—named "TheBOX" (see Supplementary Notes 7.11) and a version optimized for microscopy training courses "CourseBOX". It is supported by continuously improving documentation with step-by-step guides and tutorials. We tested the system at various conferences, workshops, and educational environments (see Supplementary Notes 9) and obtained plenty of constructive feedback to further improve the system. We noted a declining usage- and understanding barrier of new workshop-participants during these iterations due to improvements in documentation and steadily increased robustness of the cubes.

**Multimodal: fluorescence and label-free imaging.** Although we were able to show that fluorescence imaging is possible using the UC2 incubator-enclosed and light sheet configuration (e.g. fluorescence overlay in Fig. 3c), the sensitivity of the Raspberry Pi camera suffered from high noise contribution as quantified in Supplementary Notes 7.5 and the reduced sensitivity due to the Bayer pattern. Replacing the RGB Raspberry Pi camera with a cellphone featuring a back-illuminated monochromatic camera (P20 Pro, Huawei, China), capturing up to 4× more photons improved the imaging performance significantly. A quantitative comparison was obtained by acquiring mCLING-ATTO 647N (SYSY, Germany) labeled E. coli using a UC2 laser-based infinity optics fluorescence microscope (×100, $NA = 1.25$ oil, $\lambda_{exc} = 635/637$ nm, see Supplementary Notes 7.4) equipped with a Raspberry Pi or cellphone camera with a standard research-grade microscope (Zeiss Axiovert TV, ×100, $NA = 1.46$) in Fig. 3i–k. The cellphone camera clearly resolved the plasma membrane of the bacteria (see Fig. 3i–k, small sub ROI). We determined the practical resolution to be $d_{cellphone} = 0.6\,\mu m$ compared to $d_{Raspi} = 1.13\,\mu m$ and $d_{Zeiss} = 0.27\,\mu m$, at similar experimental conditions (e.g. exposure time, gain, laser intensity) using Fourier ring correlation (FRC)[38] (further quantified in Supplementary Notes 7.4). Using the GUI on the Raspberry Pi, we were further able to schedule a time-lapse series of moving fixed but mobile (e.g. in aqueous suspension) E. coli bacteria at 1 fps using the previously mentioned infinity-corrected setup (see Supplementary Video 7).

UC2 also enables the creation of more sophisticated systems. As an example, we present the creation of an image scanning microscope (ISM)[39], where we replaced the excitation laser in the previous infinity-corrected setup with a customized module hosting a laser-scanning video-projector (Sony MP.CL1A, Japan; Supplementary Notes 7.9). We compare images of GFP-labeled Human Pulmonary Microvascular Endothelial Cells (HPMEC) acquired with the UC2-ISM (Optika, ×20, $NA = 0.4$, N-plan, further information Supplementary Notes 7.9) to a state-of-the-art laser-scanning confocal microscope (Leica TCS SP5, Fluotar ×20, $NA = 0.5$, Germany) in Fig. 3e–h. The computationally

reconstructed "superconfocal" image[40] Fig. 3g shows optical sectioning compared to the wide-field equivalent Fig. 3f.

Further, when using an LED matrix (Adafruit #1487, NY, USA) as a light-source in transmission mode, the selection of the illumination wavelength, particular patterns for contrast-maximization[41] using the openKoehler module (Supplementary Notes 7.10), dark-field illumination (Fig. 3n) or quantitative phase-methods like "(quantitative) differential phase contrast" (qDPC[26], see Supplementary Notes 7.6) and "Fourier Ptychography Microscopy" (FPM[42]) are straightforward. We replaced the matrix with an LED ring (Adafruit#1463) to demonstrate computational refocussing of a recovered phase map of cheek cells (Fig. 3m) to apply "Annular Intensity Diffraction Tomography" (aIDT[43], see also Supplementary Notes 7.6 and the reconstructed Z-stack in Supplementary Video 3).

## Discussion

We here introduce a modular toolbox with the potential to serve as the truly open standard. This standard is defined by the dimensions and shape of the basic cube based on a variety of parameters and experiences to be as generic as possible. Our aim to not only create new parts, but define a common interface for the ever-growing variety of different components, was achieved. By interfacing UC2 also with existing railings or cage systems from Thorlabs, Newport, Edmund Scientific, and the like as well as with existing lab equipment, we facilitate users to start interfacing and reusing existing components and setups, therefore reinforcing the idea as an open standard.

We demonstrated the inherent versatility of the UC2 toolbox by first realizing a whole microscope life-cycle in a few steps for an incubator-enclosed bright-field configuration and then presented examples of how exchanging a few components can implement different modern microscopic techniques.

Yet there are of course also limits with respect to the long-term stability of 3D-printed setups, which are attributed to the PLA and ABS materials which deform in dependence of temperature. The iterative design-process resulted in a replaceable mechanical module with minimal bending which can be actively supported by an autofocus-routine or manual refocus. This allowed us to achieve long-term stability in multiple experiments imaging with 4 incubator-enclosed microscopes over 7 days without notable focus-drift, where in-vitro macrophage differentiation was continually observed. Access to long-term measurements allowed the replication of data published by Xia et al.[37], where the elongated shape of macrophages is correlated to their movement. The incubator-enclosed microscope proved the benefits of its inherent small footprint and high throughput capability by parallelizing experiments on a very low budget while providing customized imaging tools for e.g. microfluidic chips or inside high-safety biological environments (BSL3+) at the same time.

Another major limiting factor in fluorescence imaging (e.g. light sheet setup) is the performance of the Raspberry Pi camera used (v2.1), which can be improved with more sensitive camera sensors, e.g. from mobile phones or industrial cameras. Therefore, the light sheet system is more of a low-cost (≈400 Euro) proof-of-concept, which provides valuable insight into the method for educational users, rather than being a productive imaging tool.

With "TheBox" we introduced a sophisticated toolset for educational purposes. Together with a series of ready-to-use documentations, optical concepts (interference, image formation, etc.), and a variety of light microscopy methods we provided an openly accessible microscopic platform for a price between 100 and 600 Euro. This gives students and end-users the possibility to experience how advanced optical methods work and promote interdisciplinary approaches where several educational

topics are treated at once. Exemplary teaching material is given in Supplementary Fig. 3.

The UC2 toolbox can be easily integrated into existing frameworks like the Openflexure stage[16], Micro-manager[44], and ImJoy[45] due to its inherent modularity on both the hardware as well as on the software side. Furthermore, the existing pool of ready-to-use modules enables rapid prototyping in optics, education, and other fields. A versatile, flexible, extendable enabling tool is dearly needed in optics. With UC2 we hope to create an optical equivalent to what the Arduino represents for electronics and Fiji[46] for image processing of biomedical data, by making state-of-the-art microscopic-techniques available to everyone. We strive to counter the reproduction-difficulties by providing step-by-step protocols on the hardware-level to retrace experiments directly. We believe that the addressed community will pick up the UC2 toolbox as a true open standard and therewith supporting simpler dissemination of laboratory research and rapid system prototyping not only in research but also in education.

## Methods

**Fabrication of the components and selection of additional parts for the incubator-enclosed microscope**. A detailed description of each individual part as well as the bill of material (BOM) can be found in our Supplementary Material (Supplementary Notes 1) and is available in the GitHub-repository at http://github.com/bionanoimaging/UC2-GIT. In general, all components of the UC2 toolbox are designed using common CAD software (Autodesk Inventor 2019, MA, USA; OpenSCAD 2019.05) and were printed using off-the-shelf FDM-based 3D-printers (Prusa i3, MK3s, Czech Republic; Ultimaker 2+/3, Netherlands) where in all cases, except the Z-stage and base plate, PLA ($T_{print} = 215 °C$) was used as printing filament. The infill was chosen between 20%-40% together with a layer height of 0.15 mm which provided high enough precision and stability for all-optical setups. The monolithically printed Z-stage cube (Supplementary Notes 7.1) based on a linear or flexure-bearing and horizontally mounted baseplates for the use in the incubator were printed using ABS which provided better long-term stability at $T_{incubator} = 37°$. The Z-stage adapts to common objective lenses (i.e. RMS-thread) which gets linearly translated using a worm-drive realized with a M3 screw and nut driven by an inexpensive stepper motor (28BYJ-48, China).

The black material was used in most cases to reduce stray light or unwanted reflection and scattering. To decontaminate the printed parts, the assembled cubes were sprayed with 70% ethanol before entering the live-cell imaging lab facility (LSB2, UKJ Jena).

For the magnetic snap-fit mechanism 5 mm neodymium ball-magnets were press-fit into the printed baseplate which adapted to $M3 \times 12$ mm galvanized cylindrical screws (Würth M3 × 12, ISO 4762/DIN 912) sitting in each face of the cube to assure a stable connection. Additional wires added to the magnets and screws respectively support electro-optical modules (e.g. LED array) with electrical power (i.e. 5V, GND), where a rectifier avoided problems with the wrong polarity.

To keep the optical design simple and compact, we relied on a low-cost (15 Euro) finite corrected objective lens (×10, $NA = 0.3$, China), where the beam was folded using a cosmetic mirror (20 cents). The image, formed at a reduced tube-length ($d_{tube} = 100$ mm), was captured using a back-illuminated CMOS sensor (Raspberry Pi Camera, v2.1, UK), connected to a Raspberry Pi v3B. An additional module that incorporates a pair of motor-driven, low-cost XY micro-stages (3 Euro, $d_x = d_y = \pm1.2$ mm, Aliexpress, China) can be used (Supplementary Notes 7.1). For bright-field and quantitative imaging we used an 8 × 8 LED-array (Adafruit #1487, NY, USA) where a GUI was run on a 7-inch touchscreen (Raspberry Pi, UK), which allowed to activate LEDs individually to maximize the contrast according to Siedentopf's principle[41]. For fluorescent imaging of GFP-labeled HPMEC cells, we equipped the fluorescent module (Supplementary Notes 7.1) with two high-power LEDs in dark-field configuration (Cree, 450 nm/405 nm ± 20 nm) and added a gel color filter in front of the CMOS sensor (ROSCO #11). In-detailed information about the UC2-ISM can be found in Supplementary Notes 7.9.

**Hardware synchronization and image acquisition**. All sources together with full documentation of the software briefly described below together with an in-detail set of instructions can be found in our GitHub repository and Supplementary Notes 6.1.

A reduction of wires for "active" modules (e.g equipped with motors, LED's) was achieved by a microcontroller connecting to a wired $I^2C$-BUS (Arduino Nano, Italy) or a wireless MQTT protocol-based (ESP32 WROOM, China) network. As a master device for the 4-wired $I^2C$ connection, we choose the Raspberry Pi v3B. The ESP32 can be controlled with any MQTT-device, e.g. Raspberry Pi, cellphone, or other ESP32/Arduino microcontrollers in the same network permitting to control the device remotely (e.g. from the office).

A user-friendly Python-based[47] GUI running on a 7-inch touchscreen gives access to functions like scheduling experiments, setting up imaging modalities (e.g. illumination pattern), and hardware-/frame synchronization for several applications (e.g. incubator-enclosed microscope). Frames from the camera module (Raspberry Pi, v2.1) are stored as compressed JPEG images to save memory or have the RAW non-processed Bayer pattern data written into EXIF meta-data. In cases where cellphones (e.g. P9/P20 Pro, Huawei, China) were used as imaging devices, the open-source camera APP FreeDCam (ref. [48]) was used to have full control over imaging parameters (i.e. ISO, exposure time) and access to RAW images. USB-batteries (power banks) allowed autonomous operation in rural areas over several days.

We tested live-drift-correction to account for expanding of the material by software-based autofocus (i.e. axial defocus). As a focus metric, we used a direct spatial filter (i.e. Tennengrad)[49] and a variance-based filter as image sharpness-metric (Supplementary Notes 7.3).

**Image analysis and image processing**. A customized Python[47] script handles long-term measurements (e.g. one frame-per-minute over 1 week) by binning the RAW-data and creating a preview video. Then manually a frame of reference, where the lateral sample-drift seems to have settled, and regions-of-interest (ROI) for fix image features—here: dirt on the sensor—were defined. On a second iteration, image statistics like min, max, mean, or image-sharpness, and shift, using a cross-correlation estimation for the whole image and the ROIs with respect to the reference frame, are calculated. ABS, having a large linear thermal expansion coefficient of $70 \times 10^{-6}/K$[50], tends to deform especially dominant during the one-hour heat-up phase in the incubator. Dark and corrupted frames were excluded using the statistical measures. Shifts were applied to compensate XY-drift and a stack-mean was calculated for the green channel. Only the green channel was processed further. Flat-fielding and dirt-correction were achieved by division through the mean of the whole stack after background correction to account for unequal illumination and sensor-errors (e.g. dirt, scratches). Fiji (v1.53c[46]) was used for measuring the cell size (i.e. macrophages, see Supplementary Fig. 2), the diameter was determined manually across 10 time-frames over the whole 1-week measurement of all four microscopes. In each frame, an individual cell was selected manually before the roundness-factor was computed using a customized macro.

For task-specific image processing on the cellphone directly such as the processing of the ISM measurements or frame segmentation, we used the cloud-based image processing framework ImJoy (v0.11.15[45]) for available on our GitHub repository (Supplementary Notes 6.1).

For the quantitative phase measurements based on the aIDT, we used the publicly available Matlab (2017b, The MathWorks, MA, USA) code from Li et al.[43] with small modifications according to the optical system using the cellphone microscope (see Supplementary Notes 7.6).

Possible fluctuation of Z-stacks acquired with the light sheet microscope were registered using a cross-correlation based routine before a deconvolution based on the publicly available "GenericDeconvolution" program by Heintzmann et al. (available upon request) removed out-of-focus blur.

**Sample preparation**. Peripheral blood mononuclear cells (PBMCs) were isolated from blood donated by healthy volunteer adult donors by Ficoll density centrifugation. The study and experimental protocols used therein were approved by the ethics committee of the University Hospital Jena (assigned study number 2018-1052-BO). Briefly, blood was mixed with isobuffer (phosphate-buffered saline, PBS without Ca/Mg (Gibco, Darmstadt, Germany), 2 mM Ethylenediaminetetraacetic acid (EDTA, Sigma Aldrich, Steinheim, Germany), 0.1% bovine serum albumin (BSA, Sigma Aldrich, Steinheim, Germany)) and placed on top of Biocoll (Biochrom, Merck, Germany) without mixing in a 50 ml tube. Biocoll and blood were centrifuged at $800 \times g$ for 20 min without breaks. PBMCs were transferred in a new 50 ml tube and washed twice with isobuffer. PBMCs were seeded at a density of $1\times 10^6$ cells/cm² in X-Vivo 15 medium (Lonza, Cologne, Germany) supplemented with 10% (v/v) autologous human serum, 10 ng/ml granulocyte macrophage colony-stimulating factor (GM-CSF), and 10 ng/ml macrophage colony-stimulating factor (M-CSF) (PeproTech, Hamburg, Germany) and Pen/Strep (Sigma Aldrich, Steinheim, Germany). After 1 h PBMCs were washed twice with Gibco RPMI 1640 media (Thermofisher, MA, USA) and remaining monocytes were then rinsed with X-Vivo with supplements. 16 h after isolation monocytes were washed with prewarmed (PBS, w/o $Ca/Mg$) and incubated 7 min with pre-warmed with 4 mg/ml lidocaine (Sigma Aldrich, Steinheim, Germany) and 1 mM EDTA. Detached monocytes were placed in a 15 ml tube and centrifuged 7 min by $350 \times g$. Sediment monocytes were counted and $1.5 \times 10^5$ were seeded in a 35 mm dish and rinsed with 3 ml X-Vivo 15 with supplements. After 24 h where the cells were washed once with X-Vivo 15 and the monocytes were rinsed with 3 ml fresh X-Vivo 15 with supplements and placed in the microscope. Additional protocols can be found in Supplementary Notes.

**Statistics and reproducibility**. All representative images reflect a minimum of three biological and non-biological replicates.

## Data availability

All the data responsible for producing the figures in this article are available in the Zenodo repository https://doi.org/10.5281/zenodo.4018965. All the data supporting the findings of this study are available from the corresponding authors upon reasonable request.

## Code availability

All files such as 3D printing STL and design files, Python code and a GUI for data acquisition as well as a bill of material and user guide for printing/assembly and acquisition can be found in publicly available github repositories. We host all hardware-related components in https://github.com/bionanoimaging/UC2-GIT/, assigned with the https://doi.org/10.5281/zenodo.4041339 and all software-related components in https://github.com/bionanoimaging/UC2-Software-GIT, assigned with the https://doi.org/10.5281/zenodo.4041343. The GenericDeconv program for deconvolution is available upon request.

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

## Acknowledgments

This study was supported by the Center for Sepsis Control and Care (Federal Ministry of Education and Research (BMBF), Germany, FKZ 01EO1502) and the Leibniz Science-Campus InfectoOptics Jena, which is financed by the funding line Strategic Networking of the Leibniz Association. Additionally, this work was financially supported by the Deutsche Forschungsgemeinschaft through the Cluster of Excellence "Balance of the Microverse" under Germany's Excellence Strategy—EXC 2051—Project-ID 690 390713860 and by the European Commission through Marie Skłodowska-Curie Actions (MSCA) Innovative Training Network EUROoC (Grant no. 812954) to A.S.M. The

authors want to thank the Lichtwerkstatt Jena—Open Photonics Makerspace located at the Friedrich Schiller University Jena for sharing resources and facilities for multiple workshops. We thank The Leibniz IPHT Jena e.V. for funding the project with the Innovation-fund. Human pulmonary microvascular endothelial cells transfected with eGFP (HPMEC-eGFP) were kindly provided by Dr. Lothar Koch and Andrea Deiwick of the Institute of Quantum Optics, Leibniz University Hannover. We thank Nora Mosig, Melanie Ulrich, and Tobias Vogt for their excellent technical assistance. We thank Kaspar Podgorski for hosting and the HHMI Janelia for funding the UC2 workshop at HHMI Janelia Research Farms. We also thank Xian Hu (Edna), Kay Schink, Felix Margadant, and Oddmund Bakke for organizing, funding, hosting, and preparing drosophila and MDCK samples for the workshop at Oslo University. For providing the zebrafish samples we thank Dr. Uta Naumann from Leibniz Institute on Aging—Fritz Lipmann Institute (FLI) Jena. We thank Philipp Kahn for creating the UC2 project webpage, Eda Bingöl for supporting with the filming, and witelo Jena e.V. for hosting several UC2 workshops. We thank Ronny Förster, Tomáš Čižmár, Nico Schramma, and Kyriacos Leptos for fruitful discussions. Further thanks go to Øystein Helle from the Arctic Universtity Tromsø for preparing and Patrick Then for helping with imaging the *E. coli* bacteria. R.H. acknowledges support by the Collaborative Research Center SFB 1278 (PolyTarget, project C04) funded by the Deutsche Forschungsgemeinschaft.

## Author contributions

B.D., R.L., and S.C. conceptualized the UC2 idea, B.D., R.L., S.C., B.M., R.H. and H.W. performed data curation, B.D., R.L., and S.C. contributed to formal analysis, B.D., B.M., and H.W. developed hardware components, B.D., R.L. and X.U. developed acquisition software, B.D., R.L., A.M., and R.H. organized funding acquisition, B.D., R.H. and R.L. supervised, conceived, and planned the project, designed the instrument, interpreted the data, and wrote the manuscript. All authors read and approved the final manuscript.

## Funding

## Competing interests

The authors declare no competing interests.
