## [Peer Review File · Nature Communications]

Reviewers' Comments:

Reviewer #1:

Remarks to the Author:

In their MS UC2 – A Versatile and Customizable low-cost 3D-printed Optical open Standard for microscopic imaging, Diederich et al. present an exciting collection of 3D printed parts and design options to build optical tools like microscopes in a straightforward and cost-efficient manner. From what I can tell based on the documentation provided, the results are good, and in several cases exceed the results obtained with available systems of a similar nature (of which there are many). For example, thermal stability in the incubator long-term imaging application seems like a great feature.

Overall, the design of UC2 is very well executed, and the online documentation and open-access nature of the work is outstanding. In fact, the design has been available online for some time now, and we have already done some in depth testing of several aspects of UC2 in the lab (e.g. we quite like the magnetic snap-in feature!). From this work alone (independent of having seen the MS), I am convinced that UC2 is worth publishing in a broad journal like Nature Comms, in principle.

However, as it stands, the wonderful design and online documentation is contrasted by an at best sloppy manuscript that, in my opinion, greatly undersells UC2 and the possibilities of open hardware systems like it. Specifically:

Major

1. The presentation is sloppy. Ignoring the more than 40 spelling mistakes, formatting issues and grammatical hiccups that I found before I stopped counting, I find the overall structure a bit confusing and not sufficiently to the point. Moreover, there is a great deal of hyperbole scattered across the text. As one of many examples for these above points, from the abstract:

a. Line 2: "all time" – unclear what this means

b. Line 2: "...availability... using widely available" – not very elegant phrasing

c. Line 3 – optical, not optic

d. Line 3, sentence starting "We aim to democratize microscopy, reproduce the reproduction crisis and enhance trust into science..." – pretty grandiose...

e. Next sentence: "Due to its versatility..." - what does the "it" refer to.

f. Next sentence after that: "...stress test." – I don't see any "stress test" in the MS

Some of the figures also have similar issues:

- For example, in Fig. 3 alone I can see a missing "d" in panel l and missing spaces between 20 and micron in panels f, h, i. And it looks like there might be 2 fonts at work in that figure.

Moreover, the discussion starts off grandiose again:

g. "...stop reinventing the wheel...". Really? The authors produced the nth version of a low-cost microscope based on 3D printed parts. Yes, their design has important advances over previous options, but I find the above statement oddly misplaced. Or a few sentences on: "...We proved reproducibility...". Where is that shown?

h. The supplement is riddled with spelling and formatting mistakes

Granted, each of the above individually would be a minor point, but when put together they give a rather unfortunate impression.

2. On the other hand, when digging into the details of the results section, the description is at times quite technical, with plenty of acronyms and terms that I suspect will not mean all that much to many of the people that the manuscript is presumably aimed at. I am not sure that all the technical detail is placed best in this section, in at least the subset of cases the supplement and/or online documentation seem like a better place. I would recommend keeping the results section focused on a good quantification of the actual results (see points 3 and 4 below), and instead put much of the technical methods details elsewhere as far as reasonably possible.

3. The main documentation for the system's performance seems to be in the form of a handful of example image panels and the videos. There is essentially no quantification or meaningful comparison to other existing systems (low cost or commercial). The reader is left wondering how UC2 really compares. For example, for fluorescence imaging the authors could measure the effective psf under a variety of configurations (and precisely note these configurations) so that the reader can meaningfully compare different designs/publications. I can see that SFig 13 has USAF calibration, that's a nice start. But it does not help to judge how good fluorescence images are.

4. Related to the above, I am not convinced about the performance of the light sheet system based on the presentation as it stands. Fig. 3j seems to be a 3D rendering of half a Drosophila larva, and the corresponding video also briefly shows a zebrafish. However, that is pretty much it. There is essentially no documentation of what the samples are (transgenic lines? ages?). For example, the authors note that the Drosophila larva expresses GFP. Where? In the cuticle? That seems to be the only thing showing up in the video. To me this looks like autofluorescence. Even less information is given on the zebrafish. In general, if the authors want to claim that they built a useful low cost SPIM (maybe they did, I simply cannot tell!), they need to demonstrate this much more clearly, and meaningfully compare their results to those from not-so-low-cost systems.

Minor

1. Please include line numbers
2. Affiliation 4 is missing "Germany" in the end
3. The paragraph above directly "Computational and 3D printed Microscopy" goes a bit off topic. I would suggest to either delete it entirely or to shorten it a lot.
4. Similarly, the "Computational and..." paragraph itself is quite wordy. Perhaps shorten a bit.
5. References 3, 11, 24 have garbled urls. Similarly, most links I supplementary table 1 and 2 are broken. Please check all references and links.
6. Fig. 1 BFP not defined. In fact, the term is used 13 times in the MS, and it is never defined. I suspect it is supposed to stand for back focal plane?
7. Page 7 - "we observe significant increase...". No statistical tests are shown
8. Methods under Image analysis, the $1.01 \cdot 10^{-4}$... - is that the PLA temperature coefficient? Please clarify.
9. Same paragraph, Heinzman et al reference missing.
10. SFig. 14a. Is the picture of a bike with a bag that contains the UC2 really needed?
11. Paragraph above "Educational Area.." in supplement. ..."can be found here". There is no link on the "here".

Signed: Tom Baden

Reviewer #2:

Remarks to the Author:

This paper covers the realization of a set of systems/ tools/ blocks/ components, of which some can be 3D-printed, to build a variety of microscopy techniques. Experimental studies of cell imaging by 'different imaging modalities' underline the feasibility of the developed components and imaging systems. The authors provide a 32 pages long supplement containing important details. Low-costs of the components (below €600) are stated.

The paper appears sound, is written and illustrated well with potential for improvements. Proofs for multiple claims and therefore strong evidence for its conclusions appear to be missing in the present manuscript. Accessing full details of the methodologies, obtained results, novelty, relevance and appreciating of achievements is not possible. Based on these major technical issues, the manuscript does not meet the criteria of Nature Publishing Group journal Nature

Communications in its current form.

Claims without proof (methodology):

p. 10: ‚We tested live-drift-correction to account for vibration or expanding of the material by a software based autofocus (i.e. axial defocus)‘

Conversion (or extension) of their microscope to a light-sheet capability; presented details are limited to ‚By adding a small number of components it is possible to reconfigure ...‘

Abstract: ‚Stress-test [the system] by observing macrophage cell differentiation, apoptosis and proliferation‘; apoptosis and proliferation only shown in supplementary material; ‚We even detected an apoptotic macrophage being phagocytosed by the surrounding macrophages and rarely seen division of a macrophage‘; were apoptosis and proliferation outside the stress test and more of a side-effect of the observation

p. 6: ‚An improvement is obtained using monochromatic back-lit CMOS sensors‘;

Intro: ‚Low cost‘ and ‚... This creates inexpensive microscopic imaging devices for around 100-300 Euro‘ vs. TheBox: ‚(...) the compilation costs around 600 Euro‘; it remains rather unclear in the manuscript how costs are made up

p. 2: ‚Even though guides on how to build systems like e.g. the lattice light-sheet [12] or openSPIM [13] can be found online, it still needs trained experts for system-assembly‘; openSPIM however offers a comprehensible step-by-step guide online on how to assemble the setup on an optical breadboard

Clarity and context:

Grasping the aim and the potentially clear multiple foci of the manuscript as well as the type of realization appears challenging; in the abstract: A ‚development cycle‘, ‚different microscopy techniques‘, a ‚bright field system‘ or a ‚light sheet system‘, ‚setups‘ are motivated. On p. 4 authors refer to their realization as ‚tool‘ and ‚tool box‘; ‚a generic framework‘; on p. 5 to a ‚compact device‘ for ‚imaging‘, on p. 8 authors refer to ‚a compilation of parts‘

A large variety of objectives are presented: ‚Reproducibility‘, ‚versatility‘, ‚creativity-support‘; ‚Switching to other imaging techniques‘ (p. 4);

For non expert readers outside the field, providing a certain degree of fundamentals might be desirable. Exemplarily, ‚4f-system‘ appears graphically in Fig. 1 and ‚Fourier-optical arrangements‘, ‚superconfocal‘ are mentioned; yet their relevance to the achieved results remains unclear; important details including ‚wide-field‘ and ‚light-sheet‘ microscopy, appear mandatory towards the understanding of imaging fundamentals

To be able to fully appreciate the novelty of the approaches and achievements described here, more insight into the novelty, their imaging potentials (performance) and limitations appear mandatory; a discussion of resolutions or relevant imaging quantities appreciated (p. 7 ‚to achieve a reconstruction resolution‘)

Choice of multitude of abbreviations could be detailed: ‚UC2‘: You-See-Too (abstract), „You, See, Too“ (intro), YOU.SEE.TOO (logo), ‚HPMEC‘ introduced in acknowledgements, although used several times before ‚A-IDT‘ vs. ‚aIDT‘ (page 6/7), ‚X-VIVO‘ vs. ‚X-Vivo‘ (page 10/11), ‚EDTA‘, ‚PBS‘, ‚BSA‘, ‚RPMI‘: not introduced)

Reviewer #3:

Remarks to the Author:

Thank you for the opportunity to review the UC2 manuscript. I have known of this project for some time and appreciate the very positive impact it has had within the biomedical and microscopy research community. We have adopted some of the UC2 design ideas to the development of optical systems in our laboratory. This was possible thanks to the open-source nature of the UC2 project, and high-degree of easy-to-follow information in its repository.

The manuscript itself does provide readers with a good overview of the UC2 project and its benefits. However, the manuscript is consistent in minor problems with its structure and content -

I detail some of these below.

I believe this is an outstanding project, but that the current manuscript does not do it justice. My suggestion is for the authors to carefully undergo several internal reviews before resubmitting a revision.

Below I provide some suggestions for improvement that are important for the authors to consider:

Main Text - Major Issues:

Major #1 - Results: "We performed multiple long-term measurements under conditions of high humidity (100%) and at temperatures around 37C; CO2 = 5% over 7 days taking images with 1 frame per minute repetition rate thereby continuously monitoring the morphological changes and plasticity related monocyte-differentiation." - please comment on and provide evidence that focal stability is maintained over 7 days. Please describe the software autofocus used. Do you expect for FDM parts to maintain their mechanical stability at high humidity and temperature over long periods?

Major #2 - Results: "A later introduced spiral flexure-bearing-based stage printed using Acrylnitril Butadien Styrol (ABS) improved the stability resulting in negligible focus-drift after the pre-warming phase." - please provide quantitative evidence for this statement.

Main Text - Minor issues:

Minor #1 - Abstract: "all-time accessibility" is a strange term to use. I appreciate that many of the components of UC2 systems are off-the-shelf in a physics or optical development laboratory. Still, the same may not be accurate for biologists who may end up being the primary users of UC2 systems. I suggest using the term "improved accessibility" instead.

Minor #2 - Abstract: "We aim to democratize microscopy..." - microscopy is a broad term which includes approaches that are not light-based and that UC2 cannot cover. Can the authors be more specific?

Minor #3 - Abstract: "...reproduction crisis..." - how so? Aren't commercial systems standardized? Also, given that there is more freedom on modifying a UC2 system and more homebuilt parts does it not run the risk of being less reproducible than a commercial equivalent?

Minor #4 - Abstract: "...development cycle from basic blocks..." a reader that is not acquainted with UC2 will not know what a block means, consider first describing that UC2 is based on the use of block units that contain optomechanical components and that the sequential assembly of these elements allows your to create multiple microscope designs.

Minor #5 - Abstract: the first half of the abstract is a substantial collection of buzz-words that merit some explanation of their context. The abstract should convey more easily what UC2 is, what is the novelty and the killer application. I would advise some rewrite considering this advice.

Minor #6 - Introduction: "The growing demand in biological research for spatial and temporal resolution, image volume, molecular-tracking and high-throughput coined increasingly complex and expensive light microscope aiming to resolve and track features on molecular level at small time-scales" - unsure the term "coined" is here used correctly, microscopes should be plural

Minor #7 - Introduction: "In order to circumvent the optical resolution limit established by Ernst Abbe [3], these systems need high stability and quality of optical and mechanical components hence resulting in very expensive and

complicated setups." - this sentence is redundant as it relates the same information as the previous phrase, that microscopes are increasingly complex and expensive

Minor #8 - Introduction: "The influence of open-source approaches leads to public-accessible workshop- and project documentation." - I'm unsure there is a connection between a project being open-source and the creation of workshops and good documentation. Plenty of open-source projects exist with zero documentation and no effort for giving training.

Minor #9 - Introduction: "an open standard in optics and microscopy is clearly missing" - unclear what the authors mean, there are several well-described standards for microscopy. Are the authors proposing a new open standard? If so, how, for what and how will the standard be registered.

Minor #10 - Introduction: globally, the introduction gives an interesting view of the microscopy field and the challenges that exist. It also touches on how there is an increased capacity for researchers to build hardware in a cost-efficient manner. It would benefit from explicitly setting the need UC2 intends to solve.

Minor #11 - Computational and 3D-printed Microscopy: "This creates inexpensive microscopic imaging devices for around 100-300 Euro..." - is this true for all the devices showcased in the manuscript? Please provide a supplementary table with the cost breakdown for components of these configurations to substantiate this claim. I advise authors to consider rewriting as "... imaging devices that can cost as little 300 Euro...".

Minor #12 - Computational and 3D-printed Microscopy: "any-time (Supp. Chapter 3)." - the authors likely mean Supp. Notes, as there is no Chapter 3 in the SI.

Minor #13 - Figure 1: It is unclear why Fig. 1 starts by showing a 4f design (panel a), which is then not referenced in the text. I advise the authors to either moving panel (a) to SI or making an explicit callout to it from the text. Otherwise, panels (b) and (c) are excellent and give a good sense of the modularity of the system.

Minor #14 - Results: "Modern microscopes with infinitely corrected objective lenses often follow the 4f -configuration, where lenses are aligned such that focal-planes (f) of adjacent elements coincide to limit the amount of optical aberrations, to realize telecentricity and to predict the system behaviour using Fourier-optics [40]." - this statement is disconnected from the rest of the text. What is the idea that the authors want to transmit, that the UC2 system is based on a 4f configuration? If so, can you be explicit about this point?

Minor #15 - Results: "modelling (FDM) printers" - can you replace by "modelling (FDM) 3D printers" to make it clearer to non-experts.

Minor #16 - Results: Please replace all occurrences of "Supp. Chapter 3" with "Supp. Notes"

Minor #17 - Results: "cellular-level (i.e. < 2:2m) for ~300 Euro (Fig. 2 b)-e)." - provide and reference a table with a component cost breakdown to justify this value.

Minor #18 - Fig 3: correct "Darkfiel" to "Dark-Field" in panel I).

Minor #19 - Only video 2 is referenced in the main text, often in incorrect locations that should instead point to the many other videos provided as SI.

Minor #19 - Results: "An improvement is obtained using monochromatic back-lit CMOS sensors from a cellphone camera (e.g. Huawei P20, China)" - either show quantitative evidence or rewrite phrase as "An improvement can potentially be obtained...".

Minor #20 - Discussion: "With the application of macrophages long-term imaging presented here, we addressed the simplification and barrier reduction into optical research thereby inviting curious minds from different backgrounds to interact with, find novel methods of data-acquisition or processing or to verify and test new microscopic methods." - a confusing statement, I don't fully understand what the message transmitted to the reader is.

Minor #21 - Discussion: "With this, the UC2 system strives to fill the gap of what the Arduino is for electronics and Fiji for (microscopy) image processing" - please cite Fiji

Minor #22 - Discussion: "We introduced a sophisticated tool set for educational purposes with TheBox. The compilation costs around 600 Euro including a monitor-equipped computer and is of similar quality as commercial instruments with one to two orders of magnitude higher price tags" - correct the typo in computer, please reference Supplementary Notes and S. Video 5. As before can you provide a cost breakdown to justify the 600 Euro cost claim?

Supplementary Information - Minor Issues:

S. Minor #1: SFig 1 - please add time-stamps to the timepoints shown. In the legend replace "blabbing" by "blebbing".

S. Minor #2: SFig 2 - please add time-stamps to the timepoints shown.

S. Minor #3: SMovie 1 - fix time-stamp only partially visible. Remove unneeded labels in the movie. Please add a scale bar.

S. Minor #4: All movies - please ensure that all movies showing microscopy data have a scalebar and time-stamps.

Overall, I very much like the UC2 concept and how it is making optical hardware development more accessible. I think it can be useful and certainly inspiring, but I feel the manuscript needs a full text revision before it is accepted.

Response to comments of reviewer 1

In their MS UC2 – A Versatile and Customizable low-cost 3D-printed Optical open Standard for microscopic imaging, Diederich et al. present an exciting collection of 3D printed parts and design options to build optical tools like microscopes in a straightforward and cost-efficient manner. From what I can tell based on the documentation provided, the results are good, and in several cases exceed the results obtained with available systems of a similar nature (of which there are many). For example, thermal stability in the incubator long-term imaging application seems like a great feature.

Overall, the design of UC2 is very well executed, and the online documentation and open-access nature of the work is outstanding. In fact, the design has been available online for some time now, and we have already done some in depth testing of several aspects of UC2 in the lab (e.g. we quite like the magnetic snap-in feature!). From this work alone (independent of having seen the MS), I am convinced that UC2 is worth publishing in a broad journal like Nature Comms, in principle.

However, as it stands, the wonderful design and online documentation is contrasted by an at best sloppy manuscript that, in my opinion, greatly undersells UC2 and the possibilities of open hardware systems like it.

We appreciate the reviewer's very high opinion on our work and the suggestion that our manuscript is in principle suited for publication in Nature communication. We apologize for the many shortcomings of the previous version and hope that our additional data and many changes will now raise the quality of the manuscript to the required high standard. We revised its structure, language and writing style significantly.

We are also very pleased to read that the reviewer has already tested the system outside the manuscript in her/his own laboratory. We felt similar excitement and got a lot of inspiration from the work of others like the Baden, Henriques or the Bowman lab.

Major

The presentation is sloppy. Ignoring the more than 40 spelling mistakes, formatting issues and grammatical hiccups that I found before I stopped counting, I find the overall structure a bit confusing and not sufficiently to the point. Moreover, there is a great deal of hyperbole scattered across the text. As one of many examples for these above points, from the abstract:

- a. Line 2: "all time" – unclear what this means
- b. Line 2: "...availability... using widely available" – not very elegant phrasing
- c. Line 3 – optical, not optic
- d. Line 3, sentence starting "We aim to democratize microscopy, reproduce the reproduction crisis and enhance trust into science..." – pretty grandiose...
- e. Next sentence: "Due to its versatility..." - what does the "it" refer to.
- f. Next sentence after that: "...stress test." – I don't see any "stress test" in the MS

Thank you very much for pointing out the sloppy writing style and the detailed examples. We revised the manuscript in favor of a more concise and clear structure which focusses on the three major hypotheses. We removed over-exaggerating statements and fixed linguistic errors. The individual examples of admittedly unsuitable writing style were thereby corrected as well.

With respect to statement d, we agree that our wording here and in other places of the manuscript was too grandiose and toned it down accordingly. We also changed the stress-test statement, which was originally meant to address the harsher environmental conditions inside incubators, to now read

“We tested the system at various conferences, workshops and educational environments (see Supp. Notes S3.9) and obtained plenty of constructive feedback to further improve the system.”

- For example, in Fig. 3 alone I can see a missing “d” in panel l and missing spaces between 20 and micron in panels f, h, i. And it looks like there might be 2 fonts at work in that figure.

Thank you for pointing out these issues, which are now fixed. We unified the fonts and improved the overall look of this figure.

g. “...stop reinventing the wheel...”. Really? The authors produced the nth version of a low-cost microscope based on 3D printed parts. Yes, their design has important advances over previous options, but I find the above statement oddly misplaced. Or a few sentences on: “...We proved reproducibility...”. Where is that shown?

The statement about “reinventing the wheel” was targeted towards our modular approach making the microscope a mere assembly of modular components rather than “the nth version of a low-cost microscope based on 3D printed parts”. Yet, we agree that this statement may look oddly misplaced. We hope that our new structure now avoids such misunderstandings and the ideas behind the modular approach are now presented more clearly.

We agree that the previous version did not demonstrate the reproducibility aspect clearly enough and we removed the too bold statement in favor of a less strong statement:

“We demonstrate the versatility of the toolbox by realizing a complete microscope development cycle from conceptualization via assembly to measurement and data-evaluation for two imaging modalities.”

To test the reproducibility, we performed similar experiments multiple times (e.g. 4 incubator microscopes to perform the same experiment or transporting the microscope by bicycle, setting it up and use it right away). To allow other laboratories and people to reproduce the experiments, we provide users with comprehensive step-by-step protocols to recreate the same image quality as we present in the manuscript. The fact, that we were able to reproduce biological observations of other groups made on other systems is highlighted by the statement: “We observed a significant increase of size within 5000 min observation (ANOVA with post-hoc Turkey's), in agreement with published reports [Andreesen1983]”.

h. The supplement is riddled with spelling and formatting mistakes

We apologize for the poor style in the supplementary, which was now thoroughly revised in structure, format and spelling.

2. On the other hand, when digging into the details of the results section, the description is at times quite technical, with plenty of acronyms and terms that I suspect will not mean all that much to many of the people that the manuscript is presumably aimed at. I am not sure that all the technical detail is placed best in this section, in at least the subset of cases the supplement and/or online documentation seem like a better place. I would recommend keeping the results section focused on a good quantification of the actual results (see points 3 and 4 below), and instead put much of the technical methods details elsewhere as far as reasonably possible.

We thank the reviewer for this detailed suggestion which coined us to fundamentally revise the general structure of the manuscript and focus on 3 main messages: “versatile”, “reproducible”, “open standard”. We support these points with 3 case studies: the incubator-enclosed microscope, the light-sheet microscope and multimodal imaging using multiple different methods together with a series of benchmarks.

A lot of technical detail was now shifted to the supplement or the online documentation and referred to from the main text. Yet some key aspects such as the underlying optical concept (Fourier optics) and some important technical key specifications of the toolbox are still mentioned and briefly

described in the main text. We hope that we now found a good balance between vision, experiments and technical details.

3. The main documentation for the system's performance seems to be in the form of a handful of example image panels and the videos. There is essentially no quantification or meaningful comparison to other existing systems (low cost or commercial). The reader is left wondering how UC2 really compares. For example, for fluorescence imaging the authors could measure the effective point spread function (PSF) under a variety of configurations (and precisely note these configurations) so that the reader can meaningfully compare different designs/publications. I can see that SFig 13 has USAF calibration, that's a nice start. But it does not help to judge how good fluorescence images are.

We thank the reviewer for raising this very important fundamental aspect. Indeed, the manuscript was missing a benchmark to other (commercial) systems or gold-standard methods. Therefore, we now compare fluorescent measurements of mCLING ATTO647 labelled E. coli bacteria. In order to approach the conditions of a research grade microscope (Zeiss Axiovert TV, 1.46NA, Oil, 100x) equipped with an emCCD camera, we built an inexpensive infinity-corrected laser-based fluorescence microscope (1.25 NA, Oil, 100x) using a Raspberry Pi and a monochromatic cellphone camera. We compare the results qualitatively and quantitatively by means of optical resolution in various configurations using the Fourier ring correlation (FRC). Membrane-labelled E. coli bacteria proved to be a nice benchmark sample to this aim, since their shape and appearance is well known, and can easily be compared on different systems.

4. Related to the above, I am not convinced about the performance of the light sheet system based on the presentation as it stands. Fig. 3j seems to be a 3D rendering of half a Drosophila larva, and the corresponding video also briefly shows a zebrafish. However, that is pretty much it. There is essentially no documentation of what the samples are (transgenic lines? ages?). For example, the authors note that the Drosophila larva expresses GFP. Where? In the cuticle? That seems to be the only thing showing up in the video. To me this looks like autofluorescence. Even less information is given on the zebrafish. In general, if the authors want to claim that they built a useful low cost SPIM (maybe they did, I simply cannot tell!), they need to demonstrate this much more clearly, and meaningfully compare their results to those from not-so-low-cost systems.

We are grateful for the reviewer's comments and reworked the section including more detail on the zebrafish and drosophila samples in the supplement. We now stress that we do not claim a state-of-the-art light-sheet system but rather "Another major limiting factor in fluorescence imaging (e.g. light sheet setup) is the performance of the Raspberry Pi camera used (v2.1), which can be improved with more sensitive camera sensors, e.g. from mobile phones or industrial cameras. Therefore, the light sheet system is more of a low-cost (400 Euro) proof-of-concept, which provides valuable insight into the method for educational users, rather than a productive imaging tool." The hardware of this setup can surely be further improved to also achieve scientifically relevant quality.

Minor

1. Please include line numbers

We added line numbers to the document

2. Affiliation 4 is missing "Germany" in the end

We added Germany to the affiliation.

3. The paragraph above directly "Computational and 3D printed Microscopy" goes a bit off topic. I would suggest to either delete it entirely or to shorten it a lot.

We took this very helpful comment into account when reorganizing the structure of the current manuscript and removed this section entirely.

4. Similarly, the “Computational and...” paragraph itself is quite wordy. Perhaps shorten a bit.
This section was removed (see above).

5. References 3, 11, 24 have garbled urls. Similarly, most links I supplementary table 1 and 2 are broken. Please check all references and links.

We updated the links in references 3, 11, 24 which now appear in a different order. We also updated all GitHub links with permanent links to the commit of 6/14/2020.

6. Fig. 1 BFP not defined. In fact, the term is used 13 times in the MS, and it is never defined. I suspect it is supposed to stand for back focal plane?

Indeed, the abbreviation was not introduced before. We now introduce the abbreviation before use.

7. Page 7 – “we observe significant increase... “. No statistical tests are shown

Indeed, a statistical analysis of the acquired data was not shown. We now replaced the plot in Fig. 3 with a whisker dot plot which shows the area growth of differentiating macrophages of four simultaneously running microscopes. A significant increase in size is given for the time-point 5,000 min measured with 1-way ANOVA. The legend of Fig. 3 now also states: “Statistical testing was with one-way ANOVA and Tukey's correction, * $p < 0.05$.”

9. Same paragraph, Heinzman et al reference missing.

Indeed, the reference was missing. We changed the reference to "available upon request" since the deconvolution tool will be presented in a dedicated journal publication.

10. SFig. 14a. Is the picture of a bike with a bag that contains the UC2 really needed?

We think the figure of a bike completes the picture of having a truly field portable setup which can easily be transported to remote areas outside expensive packaging systems. It also underlines the "fun"-aspect to bring boxes to schools to give students the chance to try them out.

11. Paragraph above “Educational Area..” in supplement. ...”can be found here”. There is no link on the “here”.

We now added the missing link.

Response to comments of Reviewer 2

This paper covers the realization of a set of systems/ tools/ blocks/ components, of which some can be 3D-printed, to build a variety of microscopy techniques. Experimental studies of cell imaging by 'different imaging modalities' underline the feasibility of the developed components and imaging systems. The authors provide a 32 pages long supplement containing important details. Low-costs of the components (below €600) are stated.

The paper appears sound, is written and illustrated well with potential for improvements. Proofs for multiple claims and therefore strong evidence for its conclusions appear to be missing in the present manuscript. Accessing full details of the methodologies, obtained results, novelty, relevance and appreciating of achievements is not possible. Based on these major technical issues, the manuscript does not meet the criteria of Nature Publishing Group journal Nature Communications in its current form.

We kindly thank the reviewer for her/his critical feedback on our work. We agree that the first submission manuscript lacked a comprehensive overview of the evidence for the assertions made in the form of sustainable measurements. By thoroughly restructuring the manuscript and carrying out additional experiments, the reader should now have gained easier access to all the details of the methods.

Claims without proof (methodology)

p. 10: ,We tested live-drift-correction to account for vibration or expanding of the material by a software based autofocus (i.e. axial defocus)'

Thank you for pointing out the confusion. The previous version was admittedly confusing regarding this point. For the long-term experiments, we were able to use ABS-printed Z-stages, which, in contrast to the PLA-printed Z-stages, showed much less drift/material deformation. This obviated the need for drift correction.

However, we chose to still describe the autofocus in the supplement and characterized it with a long-term drift/defocus measurement as this may interesting for other applications. These drift characterizations also helped us to optimize the design.

Conversion (or extension) of their microscope to a light-sheet capability; presented details are limited to 'By adding a small number of components it is possible to reconfigure ...'

Sorry.

We now added more detail to the description: "In this section, we demonstrate versatility by reassembling the BF-system of the previous chapter into a light sheet microscope (Fig. 2g) by exchanging the LED array in favour of a laser-pointer, adding a second microscope-objective, beam-expander, cylindrical lens and the sample-stage using a larger baseplate. A video explaining the conversion together with a detailed conversion recipe and a detailed scheme of the openSPIM-inspired [16] setup is given in the Supp. Video S5) and Supp. Notes S7.7 respectively" Further, we hope that the reader will find our Supplementary Video 5 helpful. All the technical specifications are now also stated in the Supplemental Notes, Table S6 and S7. Additionally, we give an in-detail protocol to convert the incubator-enclosed microscope into a light sheet microscope in the Supplementary S 7.7.

Abstract: ,Stress-test [the system] by observing macrophage cell differentiation, apoptosis and proliferation'; apoptosis and proliferation only shown in Supplementary Material; ,We even detected an apoptotic macrophage being phagocytosed by the surrounding macrophages and rarely seen

division of a macrophage'; were apoptosis and proliferation outside the stress test and more of a side-effect of the observation

To address the reviewer's excellent suggestion, we restructured the content of the abstract and main part of the manuscript to be clearer regarding this claim. With the experiment on monocyte migration and shape features, we aim to show that UC2 can reproduce scientific results by means of macrophages growth: "We observed a significant increase of size within 5000 min observation (ANOVA with post-hoc Turkey's) , in agreement with published reports [Andreesen1983]". The observations of cell death were rare side-effects we were able to observe. We also removed the term "stress-test" as this is not a well enough defined term (see reviewer 1).

p. 6: ,An improvement is obtained using monochromatic back-lit CMOS sensors'; (see reviewer 1) Indeed, the manuscript was missing a benchmark to other (commercial) systems or gold-standard methods. Therefore, we now compare fluorescent measurements of mCLING ATTO647 labelled e. coli bacteria imaged using two systems. In order to approximate experimental conditions similar to a research grade microscope (Zeiss Axiovert TV, 1.46NA, 100x), we built an infinity-corrected laser-based fluorescence microscope (1.25 NA, 100x), where we compared the Raspberry Pi and monochromatic cellphone camera to the emCCD camera of the research microscope. We now present the measured optical resolution using the Fourier ring correlation (FRC). E.coli proved to be a nice benchmark sample, since their shape and appearance is well known and can be compared between systems. Additionally, we now quantify camera parameters such as readout noise and gain for the Raspberry Pi and smartphone camera. Both results can be found in the Supplementary Notes. Additionally, we perform a read-noise calibration of the two camera chips, where the monochromatic CMOS sensor from the cellphone exhibits a significantly lower readout noise.

Intro: ,Low cost' and ,... This creates inexpensive microscopic imaging devices for around 100-300 Euro' vs. TheBox: '(...) the compilation costs around 600 Euro'; it remains rather unclear in the manuscript how costs are made up

We thank the reviewer for pointing out this shortcoming. In the current version we added a comprehensive material list in the Supplementary Notes in which the costs for all modules and equipment are detailed. The final price estimates for the setups described in the paper were added to the Supplementary Notes. Additionally, we provide a full list with all components, modules, and prices in our GitHub repository which we refer to in the Supplementary Information.

p. 2: ,Even though guides on how to build systems like e.g. the lattice light-sheet [12] or openSPIM [13] can be found online, it still needs trained experts for system-assembly'; openSPIM however offers a comprehensible step-by-step guide online on how to assemble the setup on an optical breadboard This is true. We removed this misleading statement.

Clarity and context

Grasping the aim and the potentially clear multiple foci of the manuscript as well as the type of realization appears challenging; in the abstract: A 'development cycle', 'different microscopy techniques', a 'bright field system' or a 'light sheet system', 'setups' are motivated. On p. 4 authors refer to their realization as 'tool' and 'tool box'; 'a generic framework'; on p. 5 to a 'compact device' for 'imaging', on p. 8 authors refer to 'a compilation of parts'

This and similar comments of other reviewers coined us to rewrite a large part of the manuscript trying to avoid ambiguous wordings. We now provide a more clearly structured narrative and limit ourselves to the 3 main foci: "versatile", "reproducible" and "open standard" and, to show-case our claims, we detail two use-cases as well as further applications in Supplementary Notes.

A large variety of objectives are presented: 'Reproducibility', 'versatility', creativity-support; 'Switching to other imaging techniques' (p. 4);

We agree that clarity is improved by focusing on few main objectives, which we now decided to do (see above).

For non expert readers outside the field, providing a certain degree of fundamentals might be desirable. Exemplarily, '4f-system' appears graphically in Fig. 1 and 'Fourier-optical arrangements', 'superconfocal' are mentioned; yet their relevance to the achieved results remains unclear; important details including 'wide-field' and 'light-sheet' microscopy, appear mandatory towards the understanding of imaging fundamentals

Indeed, a link to the theoretical aspect of the Fourier optical principle was missing.

As the 4f principle was an important design consideration, we now inserted the following text:

"Modern microscopes with infinity corrected objective lenses often follow the so called 4f - configuration (Fig. 1 a), where lenses are aligned such that focal-planes (f) of adjacent elements coincide to limit the amount of optical aberrations, to realize tele-centricity and to predict the system behavior using Fourier-optics [36]. The name 4f results from the sum of the focal distances of a simple imaging system with two adjacent lenses stacked with coinciding focal planes, leading to 2f per lens, hence 4f in total. We adapt this inherently modular design with a generic 3D-printable framework, in which individual modules (e.g. basic cubes in fig. 1 b) are arranged in such a way that the focus and image planes of successive cubes coincide."

We believe that the reader is now able to grasp our reasoning regarding why an integer grid of 50 mm is desirable more easily. In the Table S7 (widefield) and S8 (light sheet) we now name the lens type (type, manufacturer, source) of all components used and provide details of the objectives as far as they are available.

To be able to fully appreciate the novelty of the approaches and achievements described here, more insight into the novelty, their imaging potentials (performance) and limitations appear mandatory; a discussion of resolutions or relevant imaging quantities appreciated (p. 7 'to achieve a reconstruction resolution')

We agree with the reviewer. The performance was now quantified (see response to "#p. 6" above). To address other aspects of the potential, we provide two use-cases, the incubator-enclosed and light-sheet microscope, and discuss how the system can be used as a productive tool in biological labs, as well as a tool in educational areas. Additionally, we provide a discussion on the limitations, when dealing with 3D printed components, since they inherently lack long-term stability when exposed to higher temperatures in the main part and in the Supp. Notes S 7.2. We included a drift-plot in the supplementary information to exemplify this potential problem. More sophisticated setups as mentioned in the section "multimodal" give examples how far the low-cost approach could go. We hope this gives a full demonstration of the toolbox's novelty and its imaging potentials.

Choice of multitude of abbreviations could be detailed: ,UC2': You-See-Too (abstract), "You, See, Too" (intro), YOU.SEE.TOO (logo), 'HPMEC' introduced in acknowledgements, although used several times before 'A-IDT' vs. 'aIDT' (page 6/7), 'X-VIVO' vs. 'X-Vivo' (page 10/11), 'EDTA', 'PBS', 'BSA', 'RPMI': not introduced)

We thank the reviewer for mentioning the countless formal errors. We now tried to stick to single unambiguous names and added the corresponding detail for abbreviations.

Response to comments of reviewer 3

Thank you for the opportunity to review the UC2 manuscript. I have known of this project for some time and appreciate the very positive impact it has had within the biomedical and microscopy research community. We have adopted some of the UC2 design ideas to the development of optical systems in our laboratory. This was possible thanks to the open-source nature of the UC2 project, and high-degree of easy-to-follow information in its repository.

The manuscript itself does provide readers with a good overview of the UC2 project and its benefits. However, the manuscript is consistent in minor problems with its structure and content - I detail some of these below.

I believe this is an outstanding project, but that the current manuscript does not do it justice. My suggestion is for the authors to carefully undergo several internal reviews before resubmitting a revision.

Below I provide some suggestions for improvement that are important for the authors to consider:

We thank the reviewer for the very positive feedback on our work and the honest opinion about the previous manuscript shortcomings. We therefore made a very thorough revision regarding the general structure and focus of the manuscript as well as writing style. We also incorporated the suggestions made by our local colleagues.

Major

Major #1 - Results: "We performed multiple long-term measurements under conditions of high humidity (100%) and at temperatures around 37C; CO2 = 5% over 7 days taking images with 1 frame per minute repetition rate thereby continuously monitoring the morphological changes and plasticity related monocyte-differentiation." - please comment on and provide evidence that focal stability is maintained over 7 days. Please describe the software autofocus used. Do you expect for FDM parts to maintain their mechanical stability at high humidity and temperature over long periods?

Thanks for pointing this out. This was admittedly unclear. For the long-term experiments, we used ABS-printed Z-stages, which, in contrast to the PLA-printed Z-stages, show much less (if any) drift /material deformation. During the time working on our revisions, we unfortunately had no access to the biological laboratory due to it being closed because of the corona crisis. We also had access only to a PLA-capable 3D printer, which, however, allowed us to test the XYZ long term drift stability of PLA-based stages and optimize their performance. The results of these experiments and the details of the autofocus routine are now given in Supplementary Notes: "As a focus metric we used a direct spatial filter (i.e. Tennengrad) [Royer2016a] and a variance-based filter as image sharpness-metric (Supp. Notes S3.7.1.2)."

Major #2 - Results: "A later introduced spiral flexure-bearing-based stage printed using Acrylnitril Butadien Styrol (ABS) improved the stability resulting in negligible focus-drift after the pre-warming phase." - please provide quantitative evidence for this statement.

In the Supplementary Notes we give a quantitative analysis of a PLA printed Z-stage, showing both the defocusing and the drift of the stage over several days. Unfortunately, due to limited access to the biological laboratories and printing facilities, we had no access to a non-contaminated ABS-stage to provide similar data for ABS, but the long-term recordings (see supporting video), which were not using an autofocus algorithm support a sufficient long-term stability of ABS.

Minor

Minor #1 - Abstract: "all-time accessibility" is a strange term to use. I appreciate that many of the components of UC2 systems are off-the-shelf in a physics or optical development laboratory. Still, the same may not be accurate for biologists who may end up being the primary users of UC2 systems. I suggest using the term "improved accessibility" instead.

We thank the reviewer for this suggestion. However, due to the major rewrite this statement is no longer present in the manuscript.

Minor #2 - Abstract: "We aim to democratize microscopy..." - microscopy is a broad term which includes approaches that are not light-based and that UC2 cannot cover. Can the authors be more specific?

We agree that the previous wording was too unspecific. By the reorganization of the manuscript and more focusing on the 3 main point "versatile", "enabling", "open standard", the "democratization" aspect moved to the background and is not named as such. However, we revised the manuscript to make sure that our statements only concern "light microscopy".

Minor #3 - Abstract: "...reproduction crisis..." - how so? Aren't commercial systems standardized? Also, given that there is more freedom on modifying a UC2 system and more homebuilt parts does it not run the risk of being less reproducible than a commercial equivalent?

The reviewer is right in that standard microscopes are expected to have better quality control and are thus more reproducible.

The aspect we wanted to address is, however, a different aspect. We now avoid the term "reproduction crisis" and focus on the accessibility, on the documentation, as well as on the versatility aspects. We also show that the experimental reproducibility of UC2 is good. These aspects, of course, do enable many to reproduce microscopy experiments, due to the low price, almost independent of their current equipment available to them and in this sense we address the reproduction crisis, but this is now not any longer the topic in the manuscript.

We hope that the manuscript reflects our intention of the open standard and reproducibility aspect.

Minor #4 - Abstract: "...development cycle from basic blocks..." a reader that is not acquainted with UC2 will not know what a block means, consider first describing that UC2 is based on the use of block units that contain optomechanical components and that the sequential assembly of these elements allows you to create multiple microscope designs.

We revised our nomenclature in the current manuscript and define and describe the "optical building blocks" now explicitly in the supplement.

Minor #5 - Abstract: the first half of the abstract is a substantial collection of buzzwords that merit some explanation of their context. The abstract should convey more easily what UC2 is, what is the novelty and the killer application. I would advise some rewrite considering this advice.

We removed the buzz words and rewrote the abstract and large parts of the main text. The abstract stresses the versatility and the enabling nature of UC2, which in this form we believe to be the novel. We also state a few details on the cost-effective, easy to assemble incubator-enclosed microscope as the "killer application", and mention the educational light-sheet setup although the main point really is the versatility and ability to easily interface.

Minor #6 - Introduction: "The growing demand in biological research for spatial and temporal resolution, image volume, molecular-tracking and high-throughput coined increasingly complex and expensive light microscope, aiming to resolve and track features on molecular level at small time-scales" - unsure the term "coined" is here used correctly, microscopes should be plural

We have rewritten large parts of the manuscript. This part was deleted.

Minor #7 - Introduction: "In order to circumvent the optical resolution limit established by Ernst Abbe [3], these systems need high stability and quality of optical and mechanical components hence resulting in very expensive and complicated setups." - this sentence is redundant as it relates the same information as the previous phrase, that microscopes are increasingly complex and expensive. Indeed, the information is redundant. In the shortened introduction, the corresponding part now reads: "The growing demand in biological research for spatial and temporal resolution, image volume, molecular specificity, and high throughput leads to ever more complex and expensive light microscopes to meet user requirements [1, 2]."

Minor #8 - Introduction: "The influence of open-source approaches leads to public-accessible workshop- and project documentation." - I'm unsure there is a connection between a project being open-source and the creation of workshops and good documentation. Plenty of open-source projects exist with zero documentation and no effort for giving training. Indeed, a connection was missing. We rephrased the sentence to: "A number of open source projects such as the lattice light-sheet [11] or openSPIM [12] are very well documented and spawned educational workshops, attracting users to such systems and contributing to its development."

Minor #9 - Introduction: "an open standard in optics and microscopy is clearly missing" - unclear what the authors mean, there are several well-described standards for microscopy. Are the author proposing a new open standard? If so, how, for what and how will the standard be registered. We added a comment to clarify, that a common interface standard represents the ability to combine optical and mechanical parts from different manufacturers and sources. With this we hope to allow rapid prototyping for optics. The common interface standard defined by UC2 is formulated in the module developer kit (MDK). In the manuscript it is now clarified as: "Complex optical setups require a steadily growing number of optical and photo mechanical components from different manufacturers adhering to various industry standards such as the International Organization for Standardization (ISO) or Royal Microscopy Society (RMS), whose intra-compatibility is often not given. This makes it particularly hard to tailor or even reconfigure optical systems as it requires creating handcrafted adapters. What is needed, would be an open standard Chesbrough2006 permitting to easily interface between components.

One would wish for a versatile tool, which is easy to adapt to almost any imaging task at hand, without having to start from scratch with a redesign and construction. Changing from one imaging system to another could be a mere reconfiguration rather than each time a new design. Such a tool would be useful not only for research, but also immensely helpful in optics education. It would substantially reduce the effort to build a setup and allow students to actively perform system reconfigurations within minutes. This hands-on experience will lead to understanding and enable everyone to perceive optics as a playground, where many ideas can easily be tried out. To realize such a system, an open standard is paramount, as only in this way an effortless reconfiguration can be permitted without being overly restrictive to the possibilities. Luckily many great steps in this direction have already been made: [...]

With our UC2 (You. See. Too) approach, we strive to create such an open standard. We introduce a modular, easy to build optical toolbox. Relying on the concepts of matching focal planes (see below) makes UC2 particularly easy to use, flexible to reconfigure and versatile for a large range of applications. It is equipped with open-source software, open design-files and blueprints for a large variety of setups and openly accessible documentation.

Pupils and students at all level can experience optical setups using this system. UC2 enables access to modern light microscopy for a wide-spread group of users and developers by further relying only on off-the-shelf consumer-available components (Supp. Notes S3.1 and S3.8 for the Bill of Material) and thereby creating inexpensive microscopic imaging devices for around 100-400 Euro."

Minor #10 - Introduction: globally, the introduction gives an interesting view of the microscopy field and the challenges that exist. It also touches on how there is an increased capacity for researchers to build hardware in a cost-efficient manner. It would benefit from explicitly setting the need UC2 intends to solve.

With UC2 we aim to become for optics what the Arduino already is for electronics. Namely, the open source interface between different optical components to enable rapid prototyping. We hope that the current manuscript reflects this approach: "Furthermore, the existing pool of ready-to-use modules enables rapid prototyping in optics, education and other fields. Such a versatile, flexible, extendable enabling tool is dearly needed in optics setups. With UC2 we hope to create an optical equivalent to what the Arduino represents for electronics and Fiji (Schindelin2012) for image processing by making state-of-the-art microscopic-techniques available to everyone."

Minor #11 - Computational and 3D-printed Microscopy: "This creates inexpensive microscopic imaging devices for around 100-300 Euro..." - is this true for all the devices showcased in the manuscript? Please provide a supplementary table with the cost breakdown for components of these configurations to substantiate this claim. I advise authors to consider rewriting as "... imaging devices that can cost as little 300 Euro..."

Thanks for this very useful suggestion. We now detail the costs for all modules and setups in a comprehensive Bill of Materials. The final price estimates for the setups described in the paper were added to the Supplementary Notes and Supplementary Table S7/S8.

Minor #12 - Computational and 3D-printed Microscopy: "any-time (Supp. Chapter 3)." - the authors likely mean Supp. Notes, as there is no Chapter 3 in the SI.

We indeed meant the Supp. Notes and changed it accordingly.

Minor #13 - Figure 1: It is unclear why Fig. 1 starts by showing a 4f design (panel a), which is then not referenced in the text. I advise the authors to either moving panel (a) to SI or making an explicit callout to it from the text. Otherwise, panels (b) and (c) are excellent and give a good sense of the modularity of the system.

We address this suggestion by giving a more detailed information about the Fourier optical principle and refer to the Fig 1.a) : "Modern microscopes with infinity corrected objective lenses often follow the so called 4f - configuration (Fig. 1 a), where lenses are aligned such that focal-planes (f) of adjacent elements coincide to limit the amount of optical aberrations, to realize tele-centricity and to predict the system behavior using Fourier-optics [36]."

Minor #14 - Results: "Modern microscopes with infinitely corrected objective lenses often follow the 4f -configuration, where lenses are aligned such that focal-planes (f) of adjacent elements coincide to limit the amount of optical aberrations, to realize tele-centricity and to predict the system behavior using Fourier-optics [40]." - this statement is disconnected from the rest of the text. What is the idea that the authors want to transmit, that the UC2 system is based on a 4f configuration? If so, can you be explicit about this point?

The structure was changed accordingly, and it should now be clear that the 4f principle led to the optical building block structure of UC2:

"Modern microscopes with infinity corrected objective lenses often follow the so called 4f - configuration (Fig. 1 a), where lenses are aligned such that focal-planes (f) of adjacent elements coincide to limit the amount of optical aberrations, to realize tele-centricity and to predict the system behavior using Fourier-optics [36]. The name 4f results from the sum of the focal distances of a simple imaging system with two adjacent lenses stacked with coinciding focal planes, leading to 2f per lens, hence 4f in total. We adapt this inherently modular design with a generic 3D-printable framework, in which individual modules (i.e. optical building blocks in the form of cubes Fig. 1 b are arranged in such a way that the focus and image planes of successive cubes coincide."

Minor #15 - Results: "modelling (FDM) printers" - can you replace by "modelling (FDM) 3D printers" to make it clearer to non-experts.

We replaced the term with modelling FDM 3D printers and introduced "fused deposition modeling (FDM)" at first use.

Minor #16 - Results: Please replace all occurrences of "Supp. Chapter 3" with "Supp. Notes"

We replaced the Supp. Chapter with Supp. Notes.

Minor #17 - Results: "cellular-level (i.e. < 2:2m) for ~300 Euro (Fig. 2 b)-e)." - provide and reference a table with a component cost breakdown to justify this value.

See Minor #11.

Minor #18 - Fig 3: correct "Darkfield" to "Dark-Field" in panel I).

We corrected the wrong spelling.

Minor #19 - Only video 2 is referenced in the main text, often in incorrect locations that should instead point to the many other videos provided as SI.

Thank you for pointing out this error. It was indeed a compile error in Latex, which always displayed "2" for videos and "3" for Supp. Notes. We solved this issue in the current version.

Minor #19 - Results: "An improvement is obtained using monochromatic back-lit CMOS sensors from a cellphone camera (e.g. Huawei P20, China)" - either show quantitative evidence or rewrite phrase as "An improvement can potentially be obtained...".

We thank the reviewer for this comment. To support our statement, we added a qualitative and quantitative comparison between the Raspberry Pi, the monochromatic Cellphone Camera (Huawei P20 Pro) and a standard research microscope equipped with an emCCD camera. We measured the Fourier Ring Correlation (FRC) and estimated the gain as well as the readout noise of the compact camera modules to estimate its performance. Our results can be found in the Supplementary Notes.

Minor #20 - Discussion: "With the application of macrophages long-term imaging presented here, we addressed the simplification and barrier reduction into optical research thereby inviting curious minds from different backgrounds to interact with, find novel methods of data-acquisition or processing or to verify and test new microscopic methods." - a confusing statement, I don't fully understand what the message transmitted to the reader is.

Indeed, this was a confusing statement. We have updated the biological results part and removed this sentence.

Minor #21 - Discussion: "With this, the UC2 system strives to fill the gap of what the Arduino is for electronics and Fiji for (microscopy) image processing" - please cite Fiji

We added the proper citation of Fiji.

commercial instruments with one to two orders of magnitude higher price tags" - correct the typo in computer, please reference Supplementary Notes and S. Video 5. As before can you provide a cost breakdown to justify the 600 Euro cost claim?

We summarized the prices for all setups that are listed in the manuscript. The list of parts can be found in an online spreadsheet/bill-of-material with links to the retailers. Additionally, our GitHub webpage provides an always up-to-date bill-of-material which will be wrapped in an easier online configurator soon. All details can be found in SUPP Table S7/S8.

S. Minor #1: SFig 1 - please add time-stamps to the timepoints shown. In the legend replace "blabbing" by "blebbing".

We updated the figure and the figure legend accordingly.

S. Minor #2: SFig 2 - please add time-stamps to the timepoints shown.

We updated the figure with timepoints as suggested.

S. Minor #3: SMovie 1 - fix time-stamp only partially visible. Remove unneeded labels in the movie. Please add a scale bar.

We addressed this issue by adding the suggested information such as scale-bar and timestamp to each frame and removed unnecessary entries.

S. Minor #4: All movies - please ensure that all movies showing microscopy data have a scalebar and time-stamps.

We addressed this issue by adding the suggested information such as scale-bar and timestamp to each frame and removed unnecessary entries.

We are very grateful to the reviewers for the extremely constructive and very detailed and helpful criticism, which have helped us to substantially improve the quality of our manuscript.

On behalf of all authors,

Benedict Diederich

Reviewers' Comments:

Reviewer #1:

Remarks to the Author:

The substantially revised MS is much improved. I am principally happy with it. I would however raise one points on videos:

Specifically, the videos are nice, but they are all of non-fluorescent samples. Since fluorescence microscopy is much more "useful" than non-fluorescence microscopy in the context of low cost approaches (fluorescence scopes are the ones in sparse supply in underfunded labs around the world, while almost everyone in science has access to a basic histology scope), can the authors provide a video-demo of florescence imaging? The videos are simply more helpful to judge quality compared to the stills in the MS itself. This also pertains to the light-sheet data. Can the authors please provide a video that allows clearly judging the quality of the data. This could be a 3D rendering of the final, or better still, a series of frames as the sheet swipes through the specimen.

Signed: Tom Baden

Reviewer #2:

None

Reviewer #3:

Remarks to the Author:

I thank the authors for positively and thoroughly addressing my concerns. A scale bar and timestamps still need to be added to one of the movies. I believe the article is ready for publication when this is corrected.

Response to comments of reviewer 1

The substantially revised MS is much improved. I am principally happy with it. I would however raise one point on videos: Specifically, the videos are nice, but they are all of non-fluorescent samples. Since fluorescence microscopy is much more “useful” than non-fluorescence microscopy in the context of low cost approaches (fluorescence scopes are the ones in sparse supply in underfunded labs around the world, while almost everyone in science has access to a basic histology scope), can the authors provide a video-demo of fluorescence imaging? The videos are simply more helpful to judge quality compared to the stills in the MS itself. This also pertains to the light-sheet data. Can the authors please provide a video that allows clearly judging the quality of the data. This could be a 3D rendering of the final, or better still, a series of frames as the sheet swipes through the specimen.

We thank the reviewer for this very helpful comment and understand the importance of time-series images of a fluorescently labelled sample. To connect to the rest of the manuscript, we now imaged ATTO647-labelled *E. Coli* bacteria as studied in the benchmarking of the camera sensors in a 10 minutes video (Supp. Video 7) confirming that the microscope is capable of producing high quality fluorescent time-lapse image series. We used the infinity-corrected microscope setup described in more detail in the Supplementary Notes together with the Raspberry Pi RGB camera. With more sophisticated monochromatic camera sensors, the SNR of the video can be improved. In order to clarify, that the current light sheet setup is a dedicated device for educational use only, we have added "At the moment the results show merely a proof of concept that it is possible to build a light sheet system at such a low price (~400 Euro). Better optical components and better adaptation to applications would be necessary for better performance. However, our light sheet microscope proved its usefulness in the educational area, giving the users a valuable insight in a method they frequently work with but know it only as a black box. "

Response to comments of Reviewer 3

I thank the authors for positively and thoroughly addressing my concerns. A scale bar and timestamps still need to be added to one of the movies. I believe the article is ready for publication when this is corrected.

We thank the reviewer for this very helpful comment. Indeed the scale bar and time-stamp got lost during the video-editing. We have ensured correct formatting of all videos and have added scale bars and time-stamps everywhere necessary.

We are very grateful to the reviewers for the extremely constructive, very detailed and helpful criticism, which has helped us to substantially improve the quality of our manuscript.

On behalf of all authors,

Benedict Diederich